# Prevalence of Developmental Dyslexia in Primary School Children: A Systematic Review and Meta-Analysis

**DOI:** 10.3390/brainsci12020240

**Published:** 2022-02-10

**Authors:** Liping Yang, Chunbo Li, Xiumei Li, Manman Zhai, Qingqing An, You Zhang, Jing Zhao, Xuchu Weng

**Affiliations:** 1Key Laboratory of Brain, Cognition and Education Sciences, Ministry of Education, Guangzhou 510631, China; yang-psy@foxmail.com (L.Y.); 2018022967@m.scnu.edu.cn (X.L.); zhaimm@m.scnu.edu.cn (M.Z.); 2019023044@m.scnu.edu.cn (Q.A.); 2019023159@m.scnu.edu.cn (Y.Z.); 2School of Psychology, South China Normal University, Guangzhou 510631, China; 3Shanghai Key Laboratory of Psychotic Disorders, Shanghai Mental Health Center, Shanghai Jiao Tong University School of Medicine, Shanghai 200030, China; licb@smhc.org.cn; 4CAS Center for Excellence in Brain Science and Intelligence Technology (CEBSIT), Chinese Academy of Sciences, Shanghai 200031, China; 5Institute of Psychology and Behavioral Science, Shanghai Jiao Tong University, Shanghai 200030, China; 6Institute for Brain Research and Rehabilitation, South China Normal University, Guangzhou 510631, China; 7Institutes of Psychological Sciences, Hangzhou Normal University, Hangzhou 311121, China; 8Zhejiang Key Laboratory for Research in Assessment of Cognitive Impairments, Hangzhou 311121, China

**Keywords:** developmental dyslexia, prevalence, primary school children

## Abstract

Background: Developmental dyslexia (DD) is a specific learning disorder concerning reading acquisition that may has a lifelong negative impact on individuals. A reliable estimate of the prevalence of DD serves as the basis for diagnosis, intervention, and evidence-based health resource allocation and policy-making. Hence, the present meta-analysis aims to generate a reliable prevalence estimate of DD worldwide in primary school children and explore the potential variables related to that prevalence. Methods: Studies from the 1950s to June 2021 were collated using a combination of search terms related to DD and prevalence. Study quality was assessed using the STROBE guidelines according to the study design, with study heterogeneity assessed using the *I*^2^ statistic, and random-effects meta-analyses were conducted. Variations in the prevalence of DD in different subgroups were assessed via subgroup meta-analysis and meta-regression. Results: The pooled prevalence of DD was 7.10% (95% CI: 6.27–7.97%). The prevalence in boys was significantly higher than that in girls (boys: 9.22%, 95%CI, 8.07–10.44%; girls: 4.66%, 95% CI, 3.84–5.54%; *p* < 0.001), but no significant difference was found in the prevalence across different writing systems (alphabetic scripts: 7.26%, 95%CI, 5.94–8.71%; logographic scripts: 6.97%, 95%CI, 5.86–8.16%; *p* > 0.05) or across different orthographic depths (shallow: 7.13%, 95% CI, 5.23–9.30%; deep: 7.55%, 95% CI, 4.66–11.04%; *p* > 0.05). It is worth noting that most articles had small sample sizes with diverse operational definitions, making comparisons challenging. Conclusions: This study provides an estimation of worldwide DD prevalence in primary school children. The prevalence was higher in boys than in girls but was not significantly different across different writing systems.

## 1. Introduction

Developmental dyslexia (DD) is a specific impairment characterized by severe and persistent problems in the acquisition of reading skills; these problems are not caused by mental age, visual acuity problems, or inadequate schooling [1,2]. DD, also referred to as specific reading disability or specific reading disorder, is by far the most common type of learning disability, accounting for approximately 80% of all learning disabilities [3]. Due to their frustration with reading, a great number of dyslexic children are also at increased risk of academic and social problems [4]. These children often have higher reading anxiety [5,6,7], lower positive well-being [8], and experience negative attitudes [6,9].

Typically, children begin to be formally taught to read after entering primary school, and their word-reading ability reaches adult-like levels by the end of primary school [10]. Diagnosis of DD is normally achieved after a child begins structured schooling [11]. The primary school is, thus, an important point at which early literacy screening and interventions can help to identify potential reading difficulties and address risk factors [12,13]. Therefore, the present study focuses on DD in primary school children.

Dyslexia is fairly widespread but demonstrates uncertain prevalence, ranging from 5% to 17.5% [14,15], and the variability of prevalence may be related to several factors. First, different operational definitions may result in a different prevalence. The common sets of the cut-off for reading achievement are 1 and 1.5 standard deviations (SD) below the mean for the same age [16,17,18]. Second, environmental variables (e.g., regions, socioeconomic status) and other factors (e.g., grade, sub-deficit) may also influence each child’s risk of dyslexia.

Finally, it is particularly interesting to ask whether and in what way orthographic depth influences the prevalence of DD. On the one hand, logographic scripts may yield different prevalence estimates relative to alphabetic scripts. In alphabetic scripts in which the letters represent phonemes, the prevalence of DD was reported to range from 2.28% to 12.70% [19,20], even as high as 15% and 19.90% [21,22]. Unlike alphabetic scripts, logographic scripts such as Chinese have special language characteristics: (1) the smallest written units are characters representing monosyllabic morphemes; and (2) grapheme to phoneme mappings are created in an arbitrary way [23,24,25]. As logographic scripts, such as Chinese, require the memorization of picture-like characters by rote, it was previously believed that the script presented little or no difficulty in reading [26] until 1982, when Stevenson et al. [27] reported for the first time that DD did exist among Chinese and Japanese readers. On the other hand, even within alphabetic writing systems, such systems differ in terms of orthographic depths. According to the orthographic depth hypothesis (ODH) [28], shallower orthographies are easier to learn than deeper ones. For children, it is easier to learn how to map letters onto phonological forms that are known from speech in the shallower orthographies, where in units in the written language reliably correspond to units in the spoken language. In contrast, the other two theories (the psycholinguistic grain size theory and the grapholinguistic equilibrium hypothesis) propose that the incidence of DD will be very similar across both consistent and inconsistent orthographies but that its manifestation might differ according to orthographic consistency [29,30].

In addition, the gender ratio of DD is the subject of an ongoing debate [31,32,33]. Most studies reported that more boys suffered from DD than girls, and the gender ratio of boys to girls was about 3:1 [34,35,36], but some studies found no differences in the prevalence of DD between boys and girls [18,31]. The latter interpreted the over-representation of boys in DD prevalence to be a result of bias in behavioral observation [37]. To address this problem, we conducted a subgroup analysis of gender prevalence.

Taken together, a large number of previous studies have assessed the prevalence of DD in primary school children, but the results are largely mixed. More importantly, the previous review articles did not thoroughly discuss the prevalence of Chinese DD [14,15], although the number of Chinese users is large and widely distributed. Therefore, it is necessary to include Chinese for meta-analysis.

The present study thus aimed to conduct a systematic and meta-analytical review of previous studies that reported the prevalence of DD in children in primary school. More specifically, the present study aimed to address two issues: (a) what is the prevalence of childhood DD worldwide; and (b) whether the prevalence of DD varies according to gender, writing system, and other variables.

## 2. Materials and Methods

### 2.1. Search Strategy and Selection Criteria

This systematic review and meta-analysis was conducted in accordance with the preferred reporting items for systematic reviews and meta-analyses (PRISMA) reporting guidelines [38]. The protocol of this study was registered in PROSPERO (registration number: CRD42021232958).

Looking at studies from the 1950s to 10 June 2021, two researchers (X.L. and M.Z.) independently conducted a literature search of the China National Knowledge Infrastructure, Wanfang, CQ-VIP, the China Hospital Knowledge Database, EBSCO host, ProQuest, PubMed, Web of Science, the OATD database, Cochrane, Springerlink and EMBASE, using a combination of search terms related to DD (dyslexia, reading disability, reading disorder, or learning disability), and prevalence (prevalence, detectable rate, incidence rate, or epidemiology). Then, a search of the reference lists of the studies included in the first step was performed to complement our database searches. No language or time restrictions were applied. The full search strategies for different bibliographic databases are presented in Table A1.

The study inclusion criteria were that: (i) participants consisted of primary school students (age range: 6–13 years; grade range: 1st–6th); (ii) subjects were recruited through probability sampling methods; (iii) studies included DD prevalence as a main or secondary outcome; (iv) measures with good psychometrics were used to assess the symptoms of DD; (v) no restrictions in terms of languages and published periods. For studies involving both adolescents and primary school children, the data of the primary group had to be able to be disaggregated. For multiple articles that used data from the same investigation (duplicates), only the articles with the most comprehensive results or the largest sample size were kept.

The following studies were excluded: (i) those including non-primary school students as participants; (ii) case-control studies, randomized clinical trials, review articles, and editorials; (iii) gray literature-material published by governments, organizations, and industrial or commercial entities for non-academic purposes, conference proceedings, and abstracts; (iv) no reports on DD prevalence were included in the articles; (v) studies were of specific sub-populations of participants (e.g., participants with acute or chronic disease); (vi) the articles could not be retrieved in full-text form through online databases, via library requests or email correspondence with the authors of the studies; (vii) the articles provided insufficient data regarding sample information.

After removing duplicates from different bibliographic databases, the two researchers (X.L. and M.Z.) independently screened the titles and abstracts of all retrieved records from the literature search. Then, the same two researchers assessed the eligibility of potentially relevant articles in the full text against the selection criteria. A consensus was reached for any disagreements through discussion, or the matter was decided by the other two researchers (L.Y. and J.Z.).

### 2.2. Data Extraction and Quality Assessment

Data were independently extracted from the included articles by two researchers (Q.A. and Y.Z.). The collected information included title, first author, year of publication, country, study design, sampling strategy, diagnostic materials, diagnostic criteria, sample size, the number of participants screened as DD, and prevalence estimate. The regions of study location were designated as African Region, Region of the Americas, Southeast Asia Region, European Region, Eastern Mediterranean Region, and Western Pacific Region according to the World Health Organization (WHO) criteria and as high-income countries and low- and middle-income countries according to the World Bank (WB) criteria.

We rated the quality of included articles according to the Strengthening the Reporting of Observational Studies in Epidemiology (STROBE) reporting guideline in several dimensions: sample population, sample size, participation rate, outcome assessment, and analytical methods (Table A2) [39].

### 2.3. Overall Pooled Prevalence of DD

Before pooling the prevalence estimates, the variance of raw prevalence from each included study was stabilized, using the Freeman–Tukey double arc-sine transformation [40]. All estimates were presented after back transformation. We assessed the heterogeneity of prevalence estimates among studies using the Cochran Q test and *I*^2^ index [41,42]. For the Cochran Q test, *p* < 0.05 represented significant heterogeneity. For the *I*^2^ index, values of 25% or lower corresponded to low degrees of heterogeneity, 26% to 50%, to moderate degrees of heterogeneity, and values greater than 50% to high degrees of heterogeneity [41,42].

Because of high heterogeneity (as expected and observed), a random-effect meta-analysis (following the DerSimonian and Laird method) was used to calculate the overall pooled prevalence of DD with 95% CIs throughout this study [40]. To examine whether single studies had a disproportionally excessive influence, we applied a “leave-1-out” sensitivity analysis for each meta-analysis [43]. Publication bias in the meta-analysis was detected qualitatively by a visual inspection of funnel plots and quantitatively by the Egger linear regression test and the Begg rank correlation test when more than 10 estimates were available in a single analysis [44,45,46].

### 2.4. Subgroup Meta-Analysis and Meta-Regression of DD Prevalence

We conducted subgroup meta-analyses to determine potential sources of heterogeneity. As a rule, at least three studies should be available per subgroup.

Multiple data points were generally reported in a single study. To assess the associations among various sample characteristics and the prevalence of DD, we first conducted a univariable meta-regression, if possible, followed by a multi-variable meta-regression [47]. As a rule, at least 10 data points should be available for each variable in univariable meta-regression, and 20 in multivariable meta-regression [48,49]. Data were analyzed using RStudio, version 2021.09.1-372 (R Foundation for Statistical Computing).

## 3. Results

### 3.1. Study Selection and Characteristics

As outlined in Figure 1, our initial literature search identified a total of 6564 records. After applying the eligibility criteria, a final set of 56 articles, featuring 58 studies, were included in our quantitative synthesis. A list of the 56 included articles is given in Table A3.

The detailed characteristics of the included articles can be found in Table A3. In all, 41 of the 58 studies (70.69%) reported prevalence data for both boys and girls. Of the 58 studies, 27 (46.55%) were conducted among children using alphabetic scripts, while 31 (53.45%) were conducted among children using alphabetic scripts. In addition, grade 3 was the most-studied grade (21, 36.21%) and random sampling was the most-used method (37, 63.79%), while only four studies (6.90%) had a sample size greater than 10,000. Moreover, more than half of the 58 studies (33, 56.90%) were conducted in the Western Pacific area and in middle-income countries (40, 68.97%).

### 3.2. Pooled Prevalence of DD

Table 1 illustrates the results of overall and subgroup meta-analyses. Regarding DD, the pooled prevalence was 7.10% (95% CI: 6.27–7.97%), as ascertained using random-effects meta-analysis (Figure 2).

### 3.3. Sensitivity Analysis and Publication Bias

The “leave-1-out” sensitivity analysis showed that the pooled prevalence of DD varied from 6.93% (95% CI: 6.13–7.78%) to 7.21% (95% CI: 6.38–8.09%) after removing a single study at one time (Figure A1), indicating that no individual study significantly influenced the overall pooled prevalence in the meta-analysis. Publication bias was established based on the funnel plot (Figure A2), Egger test (*t* = 6.25, *p* < 0.001), and Begg test (*z* = 1.96, *p* = 0.05).

### 3.4. Subgroup Meta-Analysis and Meta-Regression of DD

Table 1 and Figure 3 showed the prevalence of DD in different genders, writing systems, operational definitions, grades, sample sizes, sampling methods, sub-deficits, WHO regions, WB regions, and the forest plot for the difference in these factors.

There were significant differences in prevalence in terms of gender, operational definitions, and sample size. Specifically, the prevalence of DD was higher in boys (9.22%; 95% CI: 8.07–10.44%) than in girls (4.66%; 95% CI: 3.84–5.54%) (*p* < 0.001). In addition, a difference in DD prevalence was found among various operational definitions and sample sizes. The results of the post hoc analyses showed that DD prevalence was significantly lower when reporting 1.5 SD and 2SD as the cut-off values than without reporting the cut-off value (1.5 SD: 5.36%, 95% CI, 4.28–6.55%; 2 SD: 5.32%, 95% CI, 4.56–6.13%; without reporting SD: 9.10%, 95% CI, 7.18–11.21%; both *p* < 0.05, FDR-corrected). The prevalence in a large sample (more than 10,000) was significantly lower than that in smaller samples (500–1000 and 1000–1500) (10,000–: 3.13%, 95% CI, 2.32–4.06%; 500–1000: 8.43%, 95% CI, 6.83–10.18%; 1000–1500: 8.25%, 95% CI, 6.43–10.27%; both *p* = 0.09, FDR-corrected). However, there was no significant difference in the prevalence between the two smaller samples (*p* > 0.05). Univariate and multivariate regression results also showed that the subgroup of the largest sample size reported the lowest prevalence of DD.

Unexpectedly, the prevalence of DD did not differ significantly when it was stratified according to writing system (alphabetic scripts: 7.26%, 95% CI, 5.94–8.71%; logographic scripts: 6.97%, 95% CI, 5.86–8.16%; *p* > 0.05), or orthographic depth (shallow: 7.13%, 95% CI, 5.23–9.30%; deep: 7.55%, 95% CI, 4.66–11.04%; *p* > 0.05), or grade (grade 1: 7.59%, 95% CI, 2.65–14.72%; grade 2: 4.88%, 95% CI, 2.94–7.28%; grade 3: 6.35%, 95% CI, 4.78–8.13%; grade 4: 5.25%, 95% CI, 4.31–6.27%; grade 5: 7.44%, 95% CI, 4.59–10.90%; grade 6: 4.48%, 95% CI, 2.96–6.29%; *p* > 0.05). Similarly, there was no difference in the prevalence of DD among different subgroups of sub-deficits, sampling methods, WHO regions, and WB regions (*p* > 0.05).

## 4. Discussion

This systematic review and meta-analysis estimated the worldwide prevalence of DD in primary school children, with a prevalence of 7.10% (95% CI: 6.27–7.97%). There was a significant gender difference, and the gender ratio of boys to girls was about 2:1. However, there was no language-specific difference in the prevalence of DD. In addition, the prevalence was influenced by operational definition and sample size, but not by sub-deficits, grade, sampling method, WHO region or WB region. To our best knowledge, this is the first synthesized analysis on the prevalence of DD.

The pooled prevalence of 7.10% (95% CI: 6.27–7.97%) that is estimated in the present study is within the range of previous selective reviews, which have suggested that the prevalence of DD was in the range of 5–17.5% [14,15]. This is likely due to the similar diagnostic criteria of DD in most of the previous studies, in which DD was mainly defined as the low end of a normal distribution of word-reading ability [50]. Many disorders do not represent categories but instead the extremes on a continuous distribution that ranges from optimal outcomes to poor outcomes, with the underlying causal mechanisms being similar across the whole distribution. Essentially, most behaviorally defined disorders, including DD, are continuous disorders. In the present study, we were able to pool the prevalence of DD in children based on the available evidence, which allowed our systematic review and meta-analysis to provide a more comprehensive estimate of the prevalence of DD.

Interestingly, our calculation of the gender ratio regarding DD of boys to girls is about 2:1 (boys: 9.22%; 95% CI: 8.07–10.44%; girls: 4.66%; 95% CI: 3.84–5.54%) (*p* < 0.001). This result is consistent with previous studies that reported a higher prevalence of DD for boys than for girls [31,35,51]. One explanation for this gender difference in DD prevalence is that some teachers are more likely to refer boys for assessment as having special problems because boys are often perceived as being more disruptive than girls [52]. However, focusing on large-scale epidemiological studies that were not based on school-referred samples, Rutter and his colleagues (2007) also found that boys were more likely than girls to have a reading disability, indicating that teacher bias cannot account entirely for gender difference [53]. A similar phenomenon is also found in logographic writing systems [54,55]. Other explanations come from biological and environmental hypotheses, including genetic causes [56,57], immunological factors, perinatal complications, differences in brain functioning due to differential exposure or sensitivity to androgens [58], and differential resilience to neural insult [59]. Our current study cannot provide enough evidence to support or reject any of the above hypotheses; therefore, more studies on DD in both boys and girls are needed in the future. At the same time, the current findings suggest that teachers may need to pay more attention to boys who exhibit reading difficulties or disorders.

Another important finding is that the prevalence of DD did not differ significantly when stratified by writing system (alphabetic scripts: 7.26%, 95% CI, 5.94–8.71%; logographic scripts: 6.97%, 95% CI: 5.86–8.16%; *p* = 0.74). This is an unexpected result since logographic scripts are very distinctive (such as arbitrary mapping between the graphic and sound forms of words) relative to alphabetic scripts from the perspective of language; therefore, some experts believe that DD may be absent or rare in logographic scripts [26]. Research on DD has been initially and mainly conducted among the users of alphabetic scripts. Until the 1980s, researchers examined large samples of fifth-grade children in Japan, Taiwan, and the United States using a reading test and a battery of 10 cognitive tasks. However, the results showed that the prevalence of DD in Japan, Taiwan, and the United States was 5.4%, 7.5%, and 6.3%, respectively, suggesting that there is no significant difference in the prevalence of DD among different writing systems [27]. One explanation for this and our current findings is that the similarity in DD prevalence across different writing systems may be related to cross-cultural universality in the neurobiological and neurocognitive underpinnings of DD [15]. Some Western researchers and writers believed that Chinese characters are derived from pictographs, but this is not true. Instead, Chinese orthography is not primarily pictographic [27].

In addition, we found that DD prevalence did not differ across languages with different orthographic depths (shallow: 7.13%, 95% CI, 5.23–9.30%; deep: 7.55%, 95% CI, 4.66–11.04%; *p* > 0.05). These findings support the psycholinguistic grain size theory rather than the orthographic depth hypothesis [28,29]. When the orthography of the language is relatively shallow, readers can focus exclusively on the small psycholinguistic grain size of the phoneme. Otherwise, they will learn additional correspondences for larger orthographic units, such as syllables, rhymes, or whole words. Therefore, the prevalence of DD is very similar in both consistent and inconsistent orthographies, but its manifestations may vary according to orthographic depth.

Remarkably, operational definitions significantly affected the prevalence of DD. The present study found that studies with stricter operational definitions reported lower prevalence. Specifically, DD prevalence was significantly lower when using 1.5 SD and 2SD as the cut-off values than when not reporting SD (1.5 SD: 5.36%, 95% CI, 4.28–6.55%; 2 SD: 5.32%, 95% CI, 4.56–6.13%; without reporting SD: 9.10%, 95% CI, 7.18–11.21%; both *p* < 0.05, FDR-corrected). This finding is consistent with a recent selective review, suggesting that the prevalence depends on the severity of the reading problem—with lower rates for more severe problems [16]. Although the recognition of DD dates back over a century, no consensus has been reached regarding its diagnostic criteria. Therefore, many studies even use scores below 20% [60], scores in the bottom 10% [61], using different materials, and many other cut-offs for convenience. Essentially, all behaviorally defined disorders, including DD, are continuous disorders, and their operational definitions are found to be confusing in the current study. Perhaps now is not the time for change, with the continuous development of theoretical and empirical research; perhaps there will be a more appropriate operational definition for DD in the future.

It is worth noting that studies with more than 10,000 subjects reported a lower average prevalence of DD when compared to studies with 500–1000 and 1000–1500 subjects. By reviewing these studies, we found that the large sample-size studies have a common feature: that is, the diagnostic criteria were relatively strict. Only students who scored 1.5 or even 2 SD below the average on diagnostic tests were diagnosed as having DD [35,62,63]. Because of their strict diagnostic criteria, the prevalence was significantly lower than that of other subgroups [18,20]. Interestingly, in studies on other disorders, such as Tourette’s syndrome, epidemiological investigations also demonstrated that studies with larger sample sizes tended to report a relatively lower prevalence [64,65], although the reason is not clear.

There was no grade difference in DD prevalence. In the literature, the association between grade and DD prevalence remains unclear. Some studies reported that DD prevalence was lower in higher grades than in lower grades [66], and explained this finding with the argument that DD symptoms improve through systematic learning [14]. Several studies, however, have shown a higher DD prevalence in higher grades, relative to that observed in lower grades [67]. In addition, most studies reported no difference in DD prevalence among different grades [68,69,70]. Studies have shown that the level of reading ability in the first few years of school will continue in the following years and that the DD prevalence during schooling does not change greatly [20,37]. Most previous studies only studied the prevalence of DD in specific grades, mainly in grades 3 to 5, which makes it difficult to directly and empirically address the above issue [55,70,71]. In order to examine whether and how DD prevalence changes with progression through grades, future studies need to include all grades of elementary school and make the sample sufficiently representative. There was also no difference in the prevalence of sub-deficits. This shows that different tests and different indicators have no effect on the prevalence rate. That is, when there is a problem with accuracy, there is usually a problem with fluency or comprehension, and dyslexia shows no obvious differentiation.

As expected, we found significant heterogeneity when pooling the prevalence rates of DD. Thus, we performed sensitivity analyses, subgroup analyses, and meta-regression on many variables. After omitting each study one at a time (leave-1-out forest), the pooled prevalence of DD was shown to be robust and consistent. That is, no one study in this meta-analysis exerted a very high influence on our overall results. Under this condition, we further explored the patterns of effect sizes and heterogeneity in our data through a graphic display of heterogeneity (GOSH) plots [72] and found that all included studies had a low effect size and high heterogeneity (Figure A3). This result was consistent with the results of subgroup analysis, i.e., each subgroup had high heterogeneity (Table 1). In meta-regression, only the *p*-value of the sample size reached a significant level, which could explain the 39.56% heterogeneity (R^2^ = 39.56%). This indicates that the large variations in sample size among different studies may be an important reason for their heterogeneity. Another reason for heterogeneity may be that children were drawn from studies performed in a wide variety of countries with differing cultural, ethnic, social, and economic characteristics. In conclusion, such high heterogeneity in epidemiological meta-analysis is not unexpected. However, the results of this study should be interpreted with caution.

The strengths of this study include the comprehensive search strategies, a double review process, and stringent selection criteria. In our systematic review, we included only studies that were conducted in standard primary schools so that the generalizability of our results could be fully guaranteed. Moreover, we were able to pool the prevalence of DD in the included children based on the available evidence, which allowed our systematic review and meta-analysis to cover a broad scope regarding the prevalence of childhood DD.

Several intrinsic limitations of this study should also be acknowledged. First, the pooled prevalence of DD in the studied children might be affected by publication bias. We tried to minimize publication bias by searching for non-English literature and conference abstracts. Unfortunately, we could not completely rule out publication bias because of the observational nature of our study. Second, there were inherent disadvantages in pooling prevalence reports from disparate studies. For DD, sufficient data were available to pool the prevalence estimates. However, our subgroup analysis on the prevalence of any DD according to grade group, region group, and income group were only based on a limited number of studies that provided corresponding prevalence numbers. Third, ten variables across the included studies were systematically assessed, and only those studies with a large sample size were identified as showing a lower prevalence of DD. Previous studies [73,74] have suggested that socioeconomic factors were likely to contribute to disparities in DD prevalence rates in different subgroups. However, only high- and middle-income countries were assessed in the current study. Future studies are needed to explain the heterogeneity. More high-quality epidemiologic investigations on DD appear to be necessary, especially regarding different grades and in low-income countries.

## 5. Conclusions

This systematic review and meta-analysis is the first study to estimate the worldwide prevalence of DD. The results suggested that DD represents a considerable public health challenge worldwide (with a prevalence of 7.10%, 95% CI: 6.27–7.97%) and boys seem to be more affected than girls. There was no significant difference in the prevalence of DD either between logographic and alphabetic writing systems or between alphabetic scripts with different orthographic depths. However, a clear operational definition is urgently needed for the diagnosis of DD.

## Figures and Tables

**Figure 1 brainsci-12-00240-f001:**
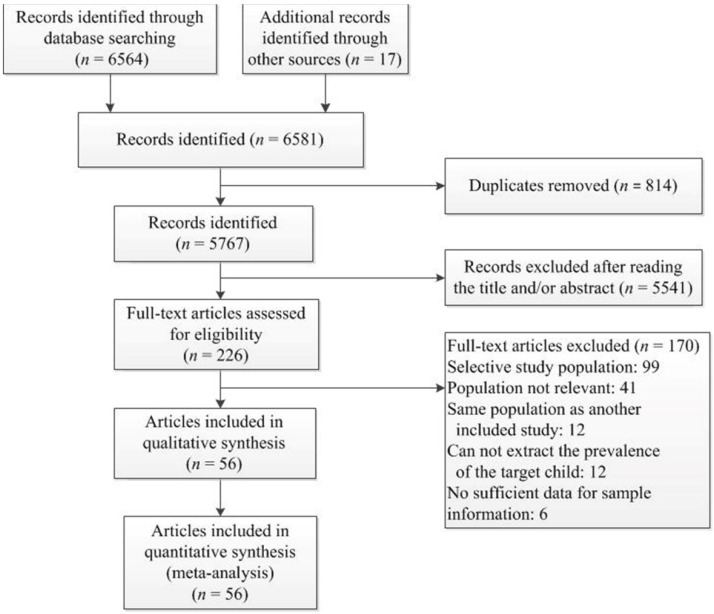
PRISMA flow diagram of literature search and study selection.

**Figure 2 brainsci-12-00240-f002:**
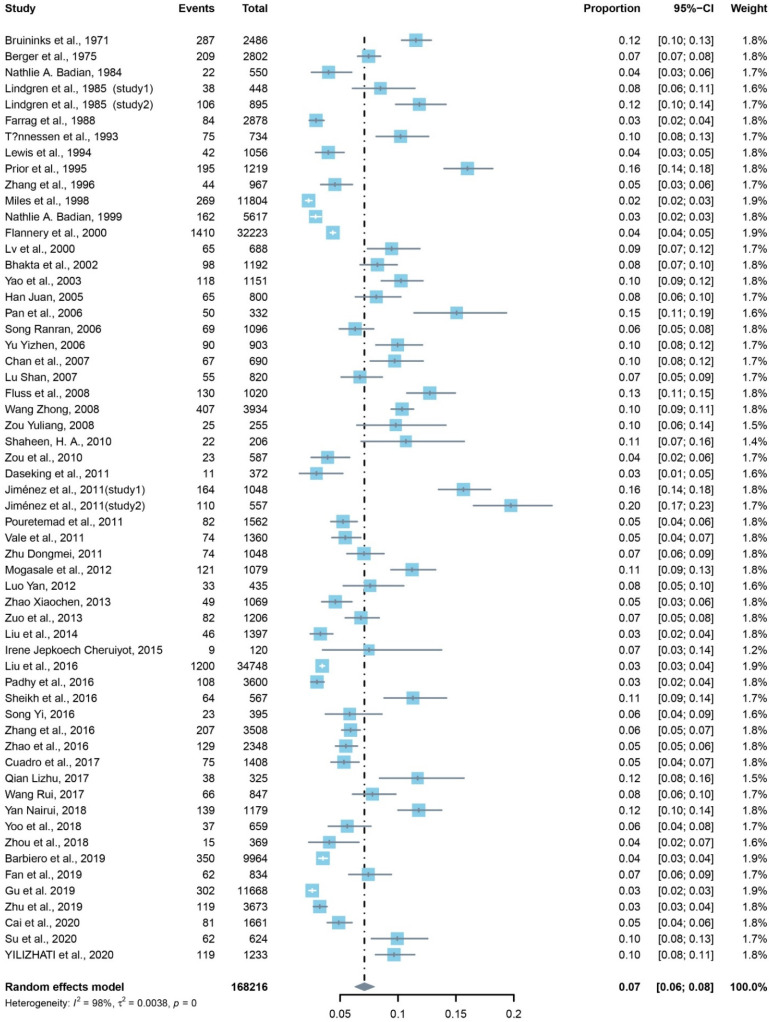
Forest plot for the prevalence of DD using random-effects meta-analysis.

**Figure 3 brainsci-12-00240-f003:**
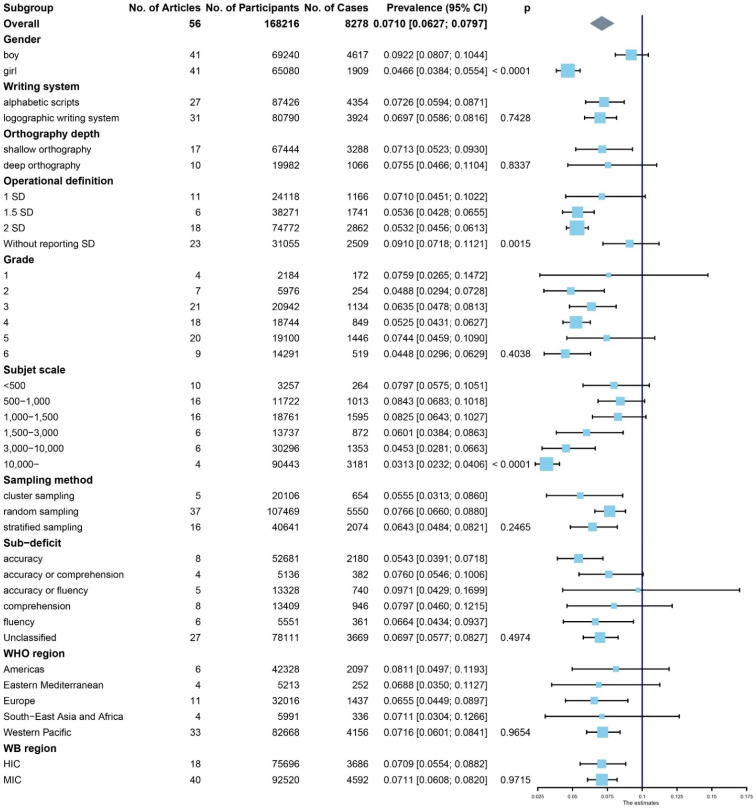
Forest plot for the subgroup meta-analysis of the prevalence of DD.

**Table 1 brainsci-12-00240-t001:** Prevalence of DD using random-effects meta-analysis and subgroup meta-analysis.

Variable	No. of Studies	Prevalence (95% CI)	*I*^2^, %	*p*-Value			
				Q Test	Egger Test	Begg Test	Subgroup Difference
Global Analysis for DD							
DD	56	7.10 [6.27; 7.97]	97.60	<0.001	<0.001	0.05	NA
Gender							
boy	41	9.22 [8.07; 10.44]	95.80	<0.001	<0.001	0.35	<0.001
girl	41	4.66 [3.84; 5.54]	95.20	<0.001	<0.001	0.17	
Writing system							
alphabetic scripts	27	7.26 [5.94; 8.71]	98.10	<0.001	<0.05	0.06	0.74
logographic writing system	31	6.97 [5.86; 8.16]	96.90	<0.001	<0.001	0.27	
Orthography depth							
shallow orthography	17	7.13 [5.23; 9.30]	98.30	<0.001	<0.05	0.19	0.83
deep orthography	10	7.55 [4.66; 11.04]	97.80	<0.001	<0.05	0.24	
Operational definition							
1 SD	11	7.10 [4.51; 10.22]	98.40	<0.001	<0.01	0.14	<0.01
1.5 SD	6	5.36 [4.28; 6.55]	87.70	<0.001	NA	NA	
2 SD	18	5.32 [4.56; 6.13]	93.70	<0.001	<0.01	0.18	
Without reporting SD	23	9.10 [7.18; 11.21]	97.20	<0.001	0.03	0.58	
Grade							
1	4	7.59 [2.65; 14.72]	96.40	<0.001	NA	NA	0.40
2	7	4.88 [2.94; 7.28]	92.00	<0.001	NA	NA	
3	21	6.35 [4.78; 8.13]	95.20	<0.001	0.06	0.15	
4	18	5.25 [4.31; 6.27]	85.00	<0.001	0.03	0.12	
5	20	7.44 [4.59; 10.90]	98.20	<0.001	0.47	0.01	
6	9	4.48 [2.96; 6.29]	93.20	<0.001	NA	NA	
Sample size							<0.001
<500	10	7.97 [5.75; 10.51]	84.00	<0.001	0.50	0.53	
500–1000	16	8.43 [6.83; 10.18]	90.90	<0.001	0.59	0.72	
1000–1500	16	8.25 [6.43; 10.27]	95.80	<0.001	0.15	0.22	
1500–3000	6	6.01 [3.84; 8.63]	97.20	<0.001	NA	NA	
3000–10,000	6	4.53 [2.81; 6.63]	98.40	<0.001	NA	NA	
10,000–	4	3.13 [2.32; 4.06]	98.10	<0.001	NA	NA	
Sampling method							
cluster sampling	5	5.55 [3.13; 8.60]	98.10	<0.001	NA	NA	0.25
random sampling	37	7.66 [6.60; 8.80]	97.20	<0.001	<0.001	0.80	
stratified sampling	16	6.43 [4.84; 8.21]	97.80	<0.001	<0.05	0.05	
Sub-deficits							
accuracy	8	5.43 [3.91; 7.18]	97.80	<0.001	NA	NA	0.50
accuracy or comprehension	4	7.60 [5.46; 10.06]	88.00	<0.001	NA	NA	
accuracy or fluency	5	9.71 [4.29; 16.99]	98.80	<0.001	NA	NA	
comprehension	8	7.97 [4.60; 12.15]	98.30	<0.001	NA	NA	
fluency	6	6.64 [4.34; 9.37]	92.40	<0.001	NA	NA	
Unclassified	27	6.97 [5.77; 8.27]	97.30	<0.001	<0.001	0.44	
WHO region							
Americas	6	8.11 [4.97; 11.93]	98.80	<0.001	NA	NA	0.97
Eastern Mediterranean	4	6.88 [3.50; 11.27]	95.90	<0.001	NA	NA	
Europe	11	6.55 [4.49; 8.97]	98.20	<0.001	<0.05	0.31	
South-East Asia and Africa	4	7.11 [3.04; 12.66]	97.50	<0.001	NA	NA	
Western Pacific	33	7.16 [6.01; 8.41]	97.30	<0.001	<0.001	0.44	
WB region							
HIC	18	7.09 [5.54; 8.82]	98.40	<0.001	<0.01	0.43	0.97
MIC	40	7.11 [6.08; 8.20]	97.00	<0.001	<0.001	0.07	

Abbreviations: WHO, World Health Organization; WB, World Bank; HIC, high-income countries; MIC, middle-income countries; NA, not applicable.

## Data Availability

All data related to the research are presented in the article.

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
