# Peer review of "Prevalence of Developmental Dyslexia in Primary School Children: A Systematic Review and Meta-Analysis"

_brainsci, 2022, doi:10.3390/brainsci12020240_

Round 1

Reviewer 1 Report

Overall the paper will add value to the literature on prevalence of dyslexia. 

Presentation can take some help from scientific writing expertise. 

Main paper is written better but the abstract needs significant work.

Few concerns to be addressed. 

Abstract: Poorly written. Starting from the very first sentence. You can state your second key finding about gender in the abstract rather just a qualitative statement.  You can also state in your abstract the details on alphabetic vs logographic language.

Last line in the abstract references to low-income countries but no analysis was referred to about this in your results or discussion.  What does low-income countries and area mean? Though limited by operational definition for dyslexia used in many studies the gender difference being significant but language differences not being significant can be phrased into your conclusion as a main clinical take away message.

Line 92, as well as in the abstract: From “inception” – please give the best start date/year you have.  Looks like around 1970.

Define primary school students  - ages and grade

Line 123 – were they the same two researchers that conducted the literature review

Why was "some population from another included study" (N=12) excluded? If it was a different, study, authors, location they should be included.

Other category - 6 were excluded. Other is not a suitable word to described or characterize in this flow chart. What was the reason for exclusion - clearly state.

Line 168 – your flow chart above says 56 studies were selected, but the denominator here is 58. Needs to be fixed. Lines 169 and - Both sentences are accounting for an incorrect number and both for alphabetic. The later is for logographic. The table is mostly reflecting 58 studies but you had only 56 studies selected  - Table s2 and Table s3

Line 206 – 209 is not clear. Rephrase. 

In the results you do not provide the findings for grade but have a paragraph on it in the discussion.

Reviewer 2 Report

General comment

The study presents a systematic review and a carefully performed meta-analysis on the prevalence of developmental dyslexia.  The PRISMA procedure as well as the statistical procedures relative to the meta-analysis appear competently carried out. The paper is generally well written and clear, and I think it will be of interest to readers of Brain Sciences.  The particular attention to the detection of dyslexia in Chinese children provides an added value to the presentation, as attention to this aspect has been relatively limited in the past.  Below I list a few comments on some parts of the text which I feel could be improved through a revision.

1. One complexity in examining the prevalence of a disturbance which lies on a continuous scale concerns the choice of an appropriate cut-off for identifying pathological performance.  In various parts of the manuscript, the authors show to be well aware of this type of problem.  In particular, I appreciate the part in which they analyze the impact of the use of different cut-offs on the various prevalence estimates obtained. 

This problem has various facets, i.e., both statistical and theoretical.  One may wonder which one is the best cut-off but, possibly, this question may not have a theoretically based response.  Indeed, some authoritative theorists (e.g., Pennington, 2006) emphasize that cut-offs for DD are “arbitrary” choices; at the very least, one could say that they are chosen on a “conventional”, not theoretically defined, basis.  Of course, saying that no ideal cut-off can be envisaged in the case of a continuous distribution, would have potentially damaging effects on the idea itself to establish prevalence rates.  I would not go that far but I wonder whether authors may elaborate and discuss at a greater extent on the bases and assumptions to carry out studies on the prevalence rates of disturbances lying on a continuous distribution.

These premises are important to interpret the results.  For example (line 232), the authors state “… we were able to pool the prevalence of DD in children based on the available evidence, which allowed our systematic review and meta-analysis to provide a more comprehensive, precise estimate of the prevalence of DD”.  One wonders what is meant with the term “precise” in this context.  I guess this concept is related to the empirical data found as it would be difficult to define a prevalence as “precise” if there is not theoretically based reason to choose a specific cut-off.  The authors may want to try clarifying this point.

2. Another point concerns the operational definition of the disturbance.  The authors (line 278) note that “Disorders like DD have a commonality, that is, there are not consistent operational definitions.  Thus, to better diagnose the disorder, a relatively consistent cut-off of reading scores is needed”. 

I find these two sentences not entirely clear and not well connected.  It is not clear to me in the first sentence what kind of conditions the authors refer to with the expression “Disorders like DD…”.  I agree that there are different operational definitions of DD, connected with different interpretations, types of orthographies, and so on.  However, from this, it does not follow to me that this can be improved by choosing a consistent cut-off.  First, operational definitions refer to the type of parameters used in the diagnosis (such as accuracy, fluency, etc.), not to statistical thresholds; second, as stated above, it is very difficult to imagine an “ideal” cut-off although a greater agreement among researchers and clinicians would certainly be useful. 
I would ask the authors to revise this part of the text.

Minor points

Abstract: Background: Developmental dyslexia (DD) during the specific impairment during the acquisition of reading skills,…

The sentence is not clear.

Line 49 “… after entering primary school. And their word reading

It should be “after entering primary school and their word reading…”

Line 68: “little or no difficulty in reading [26]. Until 1982, Stevenson et al [27] reported for the first time that DD did exist between Chinese and Japanese.”

It should read “little or no difficulty in reading [26] until 1982 when Stevenson et al [27] reported for the first time that DD did exist between Chinese and Japanese.”

Reviewer 3 Report

This review tries to examine the prevalence of dyslexia across the world. There seems nothing wrong with the technical aspects of review – i.e., collating the data, extracting the effects etc. . However, it is not clear to me what value this study has to people studying reading. The basic problem, as identified in the introduction, is that the definition and screening for dyslexia varies significantly. This happens within country, within language, within school, and even within classrooms when different people do the testing. Thus, you cannot get a fair or even consistent measurement to compare.

For example, children who read very slowly are often not identified in countries if they make few errors and this is the most common pattern of reading problem in shallow orthographies, unlike deep orthographies where children make more errors and so they are. So all we are really learning here is what we already know – you cannot learn much about the prevalence of dyslexia by only the effect sizes in studies without considering all of this. The main way people have worked out differences and rates has thus not been by  comparing reported rates across studies (which can only be done across schools in some counties – but certainly not many)  but from well controlled cross-language studies (the most famous of which is Seymour et al., 2003, although that study doesn’t deal with dyslexia).

Give the most meaningful difference in reading is language (and then SES and so on) it also makes some of the categories used here meaningless. For example, the data is broken up geographically, but it is clearly far more sensible to break it up linguistically (indeed the interaction would be useful so you could look at resources spent etc. in different countries). “Europe”, for example, could mean reading English or Greek (despite the huge difference in orthographic depth), or could mean Finland or Bulgaria (rich with a very well thought of country wide curriculum) vs. poorer. What do I learn by taking the mean here?

The same is true of character based scripts where there are differences due to linguistic differences changing the mapping between writing and speech. Korean Hangul is nothing like Hanja and Hanja is not comparable to Chinese characters.

So without any of this of analyses, I don’t see what can be learnt this. There is a list of studies which has some of this information, but this has not be used meaningfully to explore this, and it is unclear if it could be.

Round 2

Reviewer 3 Report

Please see .pdf. I have written my comments in blue.

This manuscript is a resubmission of an earlier submission. The following is a list of the peer review reports and author responses from that submission.

Round 1

Reviewer 1 Report

The manuscript that was reviewed was a proposal for a meta-analysis, rather than a publication of results per se. 

If the aim of the paper was to provide a detailed protocol to define the search strategy and analyses that would be used to investigate the topic, through meta-analytic techniques, then these should be described in much more detail, and in anticipating some of the difficulties that will be encountered in collating the secondary data.  These include, the types of sampling applied across the different samples and how these would be weighted in analyses. For example, epidemiological studies will yield very different prevalence rates than samples obtained for experiments, or via clinical environments. 

There is also potential difficulty in applying a consistent operational definition of 'dyslexia' across the different studies, was also not addressed. It appears that the authors intend to simply use face value estimates of prevalence from the literature they will sample.  This is potentially problematic because there is really no consistent definition of dyslexia that is precisely linked to specific inclusionary and exclusionary criteria.  The risk of this is that any paper that attempts to summarise a literature in which inconsistent operational definitions have been applied, risks rendering a rather meaningless summary of the data simply due to the heterogeneity of the sample. I would urge the authors to think carefully about how they would address this sampling heterogeneity in their study.  Without a more sophisticated way to to account for how dyslexia is defined, I don't think a study of its prevalence really takes the literature forward in any sufficient way.  

Reviewer 2 Report

So this looks like a request to review a PREREGISTRATION. This is interesting and the first one I have reviewed for THIS JOURNAL.

The meta-analytic techniques look sound.

The review is NEEDED and IMPORTANT.

There are ISSUES.

1) Remove "specific language impairment." This is definitely NOT dyslexia, but a speech disorder that may or may not be a precursor to dyslexia.

2) The DSM-IV created much confusion internationally by removing dyslexia and replacing it with Specific Reading Disability (315.00) with impairment in reading (F81.0).

3) The sub-deficits that must be noted  are deficits in word reading accuracy, word reading fluency, and reading comprehension. Make note as to how those deficits will be identified, noted, and categorized.

4) Make note of the criteria for 1) duration of impairment, must be present for at least 6 months in the child.

CRITICALLY: How will authors ensure that studies selected will include children that were in an adequate educational environment?

5) How will study control for presence of confounding issues such as poverty, ethic subculture difference, or structural barriers to learning, such as war, disease, or inadequate infrastructure for education? These are important for an international meta-analysis.

6) Studies must include a control or premorbid estimate of intelligence, so as to rule this out as explanation for failure to learn to read.

7) Be sure to include dyslexia severity: mild, moderate, or severe in your final rating scheme.

Specify criteria for establishing how far behind in school kids are on reading. DSM suggests 1.5 to 2.5 SD behind on standardized reading tests.

Be sure to address adequacy of test standardization for each test used to assess dyslexia in your samples.

8) Expand literature review to more convincing summarize what is known. There is NOT an adequate discussion of current meta-analyses of SRD.

Also, literature review must discuss changing terminologies for dyslexia to provide a rationale for studies included and excluded.

What are implication of your meta-analysis beyond having a better public health baseline?

Round 2

Reviewer 2 Report

The authors have made substantial clarification of their sampling, design, and methods. I feel that this should be accepted at this time.